# Barriers and Enablers of Health Service Utilisation for Childhood Skin Infections in Remote Aboriginal Communities of Western Australia

**DOI:** 10.3390/ijerph17030808

**Published:** 2020-01-28

**Authors:** David Hendrickx, Ingrid Amgarth-Duff, Asha C Bowen, Jonathan R Carapetis, Robby Chibawe, Margaret Samson, Roz Walker

**Affiliations:** 1Telethon Kids Institute, University of Western Australia, Perth, WA 6009, Australia; 2IMPACCT (Improving Palliative, Aged and Chronic Care through Clinical Research and Translation), University of Technology Sydney, Sydney, NSW 2007, Australia; 3Puntukurnu Aboriginal Medical Service Unit 5, 15, Iron Ore Parade, Newman, WA 6753, Australia; 4Jigalong Community Council, Pmb 8, Newman, WA 6753, Australia

**Keywords:** Aboriginal health, remote health, skin infections, skin sores, health service utilisation

## Abstract

In Australia, children living in remote Aboriginal communities experience high rates of skin infections and associated complications. Prompt presentation to primary care health services is crucial for early diagnosis and treatment. We performed a qualitative study in four remote Aboriginal communities in the Pilbara region of Western Australia to explore factors that affected health service utilisation for childhood skin infections in this setting. The study consisted of semistructured interviews and focus group discussions with parents and carers (*n* = 16), healthcare practitioners (*n* = 15) and other community service providers (*n* = 25). We used Andersen’s health service utilisation model as an analytical framework. Our analysis captured a wide range of barriers that may undermine timely use of health services for childhood skin infections. These included general factors that illustrate the importance of cultural competency amongst healthcare providers, patient-centred care and community engagement. Relating specifically to health service utilisation for childhood skin infections, we identified their apparent normalisation and the common use of painful benzathine penicillin G injections for their treatment as important barriers. Health service utilisation in this setting may be enhanced by improving general awareness of the significance of childhood skin infections, actively engaging parents and carers in consultation and treatment processes and strengthening community involvement in health service activities.

## 1. Introduction

Forty years after the Alma-Ata declaration, where the international community set its sights on the goal of “Health For All”, health and health service access inequities persist between and within countries [1]. The United Nations’ 2015 Sustainable Development Goals highlight the continued need for improving universal health coverage and ensuring access to quality health services for all [2]. These goals are highly relevant for Indigenous populations across the world, who continue to experience health inequity and a disproportionate burden of disease [3]. 

In Australia, Aboriginal people have a lower life expectancy, a higher infant mortality rate, are overrepresented in hospitalisation admission data and suffer from a high burden of infectious and chronic diseases [4,5,6,7]. Many Australian Aboriginal people live in rural and remote areas [8], where environmental factors and living conditions, such as overcrowding, are associated with a high prevalence of infectious diseases [9,10,11,12,13] and excessive rates of childhood ear, skin and respiratory infections [14,15,16,17,18,19]. Aboriginal people experience many difficulties in accessing primary health services [20,21,22,23,24,25], which further exacerbate their health status and undermine universal health coverage in Australia [24]. Notwithstanding the work that has already been undertaken on this topic, few studies have explored the primary health service access barriers that Aboriginal people experience in remote communities.

As part of a broader research project on the impact of childhood skin infections in remote Aboriginal communities located in Western Australia’s (WA) Pilbara region [26], we performed a qualitative study on community knowledge, attitudes and practices in relation to skin infections. Here we report specifically on our qualitative findings pertaining to barriers and enablers of health service utilisation.

## 2. Materials and Methods 

### 2.1. Study Setting

The study setting is a sparsely populated area of the remote Pilbara region, WA, Australia. This is the traditional country of the Martu, an Aboriginal people that up until the 1960s had only had intermittent contact with European Australians [27]. The study included participants from four remote Aboriginal communities (communities A to D) and a regional mining town (town E). Town E provides regional services and includes a small Aboriginal settlement on its outskirts. The Martu are a mobile people, and travel throughout the wider region and between the communities and town E is common. Driving distance from the town to the remote communities varies from 150 to 700 km, over well maintained, unsealed roads. Each of the four communities has a functional airstrip, a school (kindergarten to year 12), a local administration office and a small shop that stocks basic food and other goods. Table 1 summarizes characteristics of each of these localities.

All four communities have a clinic managed by the same Aboriginal Medical Service (AMS), which provides primary health care to Aboriginal people living in the region. At the time of the study, the clinics in the three smaller communities (B, C, D) were operated by one or two nurses at any one time. The clinic in the larger community (A) was usually serviced by two to four nurses and a permanent general practitioner (GP) who would visit the other three communities by plane once a month over a three-day period. Allied health professionals and specialist health services visit the communities on a regular schedule. Town E is not serviced by an AMS but does provide health services via a private GP clinic and a small public hospital with emergency services. The Royal Flying Doctor Service provides emergency medical care and medical evacuations to the region as required, including all four study communities [28].

### 2.2. Data Collection and Study Population

We employed an inductive qualitative research strategy to perform a series of semistructured interviews and focus group discussions (FGDs; using a yarning methodology that conforms to Aboriginal conversational norms and practices [29,30]) between October 2014 and November 2015. Given our focus on health service utilisation concerning childhood skin infections, we purposively selected three distinct groups of participants who, through their family relationships or the roles they fulfilled in community, interacted with children on a daily basis [31]. Table 2 provides a summary of our data collection activities.

Group 1 consisted of Aboriginal parents and carers who were selected through convenience sampling from community play groups, with snowball sampling to identify additional participants [31]. Group 2 were nurses employed by the AMS or other health services that operated in the region. All healthcare practitioners had at least several years of experience in the field of remote Aboriginal health and had been working in the Pilbara region for varying amounts of time, ranging from one month to many years. Group 3 were other service providers, such as school staff and community organisation workers.

We performed 34 semistructured interviews and nine FGDs (Table 2). The comparably high number of interviews in Group 3 is due to the range of community roles represented in this group. The number of participants for each FGD varied between two and four, a relatively low number to facilitate a greater depth of discussion. Data collection continued until participants from across the whole study area had been recruited and until no new themes emerged.

All interviews and FGDs were performed by either DH (male) or ID (female). To ensure cultural safety, Group 1 interviews and FGDs, with mostly Aboriginal mothers and female carers, were performed by ID. RW, a senior researcher in Aboriginal health with extensive experience in the Pilbara region provided support. The semistructured question guide covered five major topics: (1) perceived child health issues in the community; (2) knowledge, attitudes and practices around skin infections; (3) perceived underlying causes of child health issues; (4) perceived barriers and enablers to child health care provision in general and for skin infections in particular; (5) suggestions on how child health care provision in the region could be improved. Here we report our findings in relation to discussion topic 4. Our other findings are reported elsewhere [32].

### 2.3. Data Analysis

All interviews and FGDs were audio recorded and transcribed verbatim, with one exception: handwritten notes were taken during an interview with one carer who declined voice recording. During FGDs, the facilitator used notes on butchers’ paper to facilitate discussion. All transcripts were imported into QSR NVivo 10 for data analysis. 

We employed a conventional content analysis strategy to systematically identify and structure themes as they were reflected in our data [33]. The two principal investigators (DH and ID) familiarised themselves with the data through the transcription process and a rereading of the final transcripts. The coding process used the five main themes from the semistructured question guide to construct a generic coding tree. RW reviewed the coding and provided support in the interpretation of the data. Additional hierarchical levels of subthemes were added to this structure as new themes emerged from the data.

We applied Andersen’s base conceptual model of health service utilisation [34] to structure our findings. This framework is well established in the literature and its wide scope was considered to be well suited for our broad exploration of barriers and enablers [35]. The model proposes three major factors that affect access to health services: (i) predisposing characteristics, in reference to demographic, social, economic and cultural factors, including beliefs and attitudes; (ii) enabling resources, or contextual factors that influence health service utilisation, such as social relationships (e.g., family, support groups), community characteristics (e.g., availability of health services) and the health system; and (iii) need, whether perceived (by community and patients) or evaluated (based on the judgment of healthcare professionals). We examined the degree to which these factors affected health service utilisation at three distinct analytical levels: that of the client (parents, carers, children), the provider (the community clinic and its staff) and the system level (in reference to the wider health system, its values, policies and socio-economic characteristics). We created a health service utilisation matrix (see Figure 1) to systematically categorise the findings that emerged from our analytical process.

In our description of the results we present our findings in three main sections, analogous to the three chief components of the Andersen model. Within each section, we describe our findings for each of the three analytical levels of the health service utilisation matrix (client, provider, system). 

### 2.4. Ethics

Written informed consent was sought from all patient/carers, healthcare practitioners and other service providers who participated in the study. Ethics approval for the study was obtained through the Western Australian Aboriginal Health Ethics Committee (HREC 510) and the University of Western Australia Human Research Ethics Office (RA/4/1/6563). A reference group of senior Aboriginal and non-Aboriginal health researchers provided oversight during the design and implementation of the study.

## 3. Results

The interviews and FGDs provided a rich dataset for interpretation through the health service utilisation matrix lens. To adequately describe the wide range of themes elicited, we stratified our findings into three separate health service utilisation matrices (Table 3, Table 4 and Table 5), each reflecting the barriers and enablers that were elaborated by each of the three participant groups. A limited selection of quotes has been included here to illustrate several key themes that emerged from our analysis. An extensive structured selection of quotes representative of all themes documented here are provided in the Appendix A.

### 3.1. Predisposing Factors: Structural Demographic, Social, Economic and Cultural Aspects

#### 3.1.1. At the Client Level

All study participants described the impact of beliefs, norms and attitudes in the decision to seek care for skin infections. Carers spoke of a general shyness and shame around presenting to the clinic, borne out of a fear of judgement. A few healthcare practitioners and service providers also spoke on this topic. Some service providers highlighted that carers might sometimes be reluctant to take their child to the clinic for fear of the Department for Child Protection and Family Support being informed, if the healthcare practitioner were to judge that the carer was not taking appropriate care of their child. This barrier was mentioned both generally and in relation to skin infections in particular.

Shame was also discussed in relation to experiencing pain. Healthcare practitioners and service providers spoke about their perception that Martu children have a high tolerance for physical discomfort and pain. They argued that children would not easily complain about health issues—including skin infections—unless the problem became more severe, leading to delayed presentations. It was suggested that this reluctance to show pain was a cultural construct. However, several service providers suggested that Martu children may be inherently less sensitive to physical discomfort relative to their own personal experiences of pain and ill health.

Healthcare practitioners and service providers commented on their belief how differing norms around health and child rearing might affect clinic utilisation by children, their parents or carers. A commonly held belief amongst these participant groups was that the Martu place other priorities above personal health, such as family, country and cultural practice. When it came to child health, some healthcare practitioners and service providers shared their perception that some carers were not proactive enough in bringing their children to the clinic. In such cases, negative connotations around not taking responsibility and expecting children to fend for themselves were raised by participants of both these respondent groups.

Other predisposing factors discussed included a perceived lack of carer education and health literacy, social taboos (restrictions around family relationships, skin groups and gender that may affect clinic visiting behaviour), negative past experiences in the clinic (for skin infections, this was related in particular to painful benzathine penicillin G injections (BPG) administered for the treatment of skin sores), language barriers (carers not feeling confident that they could express themselves adequately in English or understand the healthcare practitioner), a possible preference in some cases to try bush medicine prior to presenting to the clinic (leading to delayed presentations), and possible apathy and disempowerment (a sense that some Martu might not feel like they have control over their lives, affecting health-seeking behaviour).


*“People get shame with the sores. You know they don’t like to show anyone they have them.”*
(carer)


*“I think that on a scale of priority, health comes at the bottom simply because it’s not anywhere different in any part of the world. If drug and alcohol issues, nutrition, family issues, domestic violence-all of those things—money issues, and grief, their position in the world, their place in our society, really that’s—most people out here would be confronted with personal life traumas as stress factors that most of us really only maybe experience every ten years or five years. […] There’s a powerlessness there where I think that people are just—there’s a kind of a, not giving up, but resigned.”*
(healthcare practitioner)


*“Ninety percent of the nurses over there they don’t understand Martu. They have got no idea what they’re saying. I took a girl over to see a doctor and the doctor had been here 15 years and I had to translate for them. There is a big major language barrier. The staff won’t break it down to simple language […] But the Martu people will say they understand. They will nod their head to the doctor and say they understand even though they don’t.”*
(other service provider)

#### 3.1.2. At the Provider Level

Several carers reported that healthcare practitioners held prejudiced views of them, leading to the fear of judgement and shame experienced by clients. Healthcare practitioners described several predisposing factors affecting their ability to deliver high quality care and appropriately interact with clients. Some described the challenges they experienced of working in small, remote communities. Physical remoteness and a sense of isolation were described as stressors, particularly in reference to the smaller communities where only one or two resident healthcare practitioners may be stationed. Long working hours, limited opportunities to prioritise preventative health and the stress of managing acute medical emergencies without back up or medical support were mentioned. One healthcare practitioner commented on the frustration they felt because of a perceived lack of progress in the communities’ health status.

Some healthcare practitioners discussed perceived training gaps. Of direct relevance to skin infections was the sense that healthcare practitioners are not always adequately trained in the administration of BPG injections and the related techniques to minimise the pain of the BPG injection, which is well established to be one of the most painful intramuscular injections to administer. Another training gap identified by healthcare practitioners concerned culturally appropriate practice, as illustrated by one nurse’s recollection of her surprise at how little cultural orientation she was provided prior to starting her role in a remote Aboriginal community. This was also reflected in several anecdotes reported by participants in each of the groups around culturally inappropriate practices of healthcare practitioners.


*“[in reference to mainstream health services in town E] The main problem is the shame and judgment going to the clinic. […] The staff at the clinic think that Aboriginal people carry disease. They think ‘oh not that black disease-carrying person again’. That’s why the Aboriginal people don’t want to go to the clinic.”*
(carer)


*“Challenging… not because of being busy, but because of remoteness, loneliness and isolation. The isolation there really is heavy because… basically you know the next doctor for your support is 900 km away. The next community is [name of other community]. It’s also in the middle of nowhere. So basically you are very far, you really feel it, you’re very far, there’s nothing, very few people to talk to. And during the festive season, like now, everyone goes over heat. It reaches up to 55 degrees there. They leave the community. So sometimes you are only two [people] in the community.”*
(healthcare practitioner)


*“The health staff are learning as they go. It is not any disrespect to the doctors or nurses […] It’s a difficult injection to give [in reference to BPG] but it depends on the nurse giving it. It just depends on your technique and training. Some of them are pretty rough up there […] The nurses don’t do the 12 months additional training where you learn the culture, learn the ways. And that is important. This is going to become un-stuck soon because they don’t know who is allowed in the clinic.”*
(other service provider)

#### 3.1.3. At the System Level

Healthcare practitioners and other community service providers discussed an apparent tension between the biomedical and traditional framing of health, and the need to reconcile both in clinical practice. Both respondent groups described through anecdotes and conceptual discussion how traditional beliefs and practices around health and healing remained important for the Martu. Most healthcare practitioners discussed the need to engage with traditional Martu health frameworks in delivering health care, although occasional dismissive comments were also noted. Some healthcare practitioners described anecdotes of how they engaged with traditional beliefs and practices, primarily for pragmatic purposes: to humour patients or to improve overall compliance, as long as it did not undermine medical treatment.


*“In some communities where there’s a lot of strong elder people that still practise Maban [powerful spiritual men], or bush medicine, I can almost guarantee you that they would’ve seen a Maban before they’ve seen me, or it’s part and parcel. They coexist, and we don’t acknowledge that enough. I know that in some other parts of Australia, it’s starting to involve it more. But in these parts here, I find that almost unbelievable that as an Aboriginal health organisation we are not encouraging Aboriginal healing side by side in our practice. Because it’s happening anyway. I know they’re highly protective of it, so it’s not just because we don’t acknowledge it. […] I think that there is another avenue for us to maintain their culture by encouraging that, so young people feel they want to go down that way. I mean, it happens irrespective of us, not that we’re training people up for that. Maybe it’s happening in a healthy way? I certainly worked with people on that level, and have let the people use the clinic for use of Maban, and I’ve seen it practised in front of me. And that again comes down to whether they trust you. Whether I believe in it or not is irrelevant. If you really want to be effective in your practices then you also have to acknowledge that that part also plays a part whether it’s placebo or does have an effect.”*
(healthcare practitioner)

### 3.2. Enabling Factors: Contextual Aspects of Social Relationships, Community Characteristics and the Health System

#### 3.2.1. At the Client Level

An important consideration in accessing health services from the carer perspective was the real or perceived costs of health care. While free care for Aboriginal people is generally available in the study area, in town E healthcare practitioners reportedly were not always aware of the publicly-funded benefits that Aboriginal people have access to (e.g., Closing the Gap funding [36]), leading to unnecessary out-of-pocket expenditure. Some carers living in Town E reported driving to the nearest remote community three hours away to access the free AMS clinic. Carers also expressed concern about the possible costs incurred if referral to tertiary health services was required. Comments by some service providers and healthcare practitioners echoed these financial issues and considerations. 

Carers discussed how their impressions of healthcare practitioners’ personalities, attitudes and professional capabilities played a role in the degree to which they would access health services in the communities. Some spoke about their lack of trust in certain healthcare practitioners’ medical knowledge and practice. Some community service providers echoed similar concerns, while others were more positive about the care they had received. 

Reflecting on further enabling factors that might affect care-seeking behaviour for childhood health issues, one service provider pointed out the need to take into consideration the parent or carers’ own health status. They described how carers may struggle with physical and mental health problems of their own, which constitute an additional barrier to taking their child to the clinic, particularly for skin infections or other health issues that might be considered relatively benign.


*“[in reference to mainstream health services in town E] Even here they do this closing the gap scheme, but all of the black fellas they don’t know that they can go there and see the doctor for free. They need to tell us this stuff you know?”*
(carer)


*“Another component to it, why things may not be as good, I think if there’s a good relationship between nursing staff or medical staff in the community, parents are more likely to bring their children, or the children come themselves at an early stage. And then also depends on how school and the health centres work together. If there’s a good working relationship and the community feels good about the health staff, they tend to come early. If they don’t then it’s left until the last moment.”*
(healthcare practitioner)


*“A lot of these people who are looking after these kids have early stage dementia or other major health problems themselves.”*
(service provider)

#### 3.2.2. At the Provider Level

From the parent and carer perspective, a key enabling factor for health service utilisation pertained to the characteristics of the clinic staff and their practices. Staff that actively established relationships with clients and demonstrated culturally secure practices were described in a positive light. Parents and carers discussed the difficulty of establishing such relationships given the high turnover of healthcare staff at AMS clinics. To illustrate the importance of relationships in the remote health care context, carers and service providers spoke of AMS healthcare practitioners who had spent extended periods of time in the communities in the past and with whom carers, children and other Aboriginal people living in the communities had established trusting, personal relationships. The need for Aboriginal Community Health Workers was also underlined by carers, pointing to their importance in relationship building and outreach activities. A lack of such activities undertaken by the AMS and its staff was highlighted as an important gap by both carers and service providers.

While some healthcare practitioners and service providers discussed the importance of culturally secure practices by medical service staff, negative experiences described by carers suggested that theory and practice are not necessarily aligned. Carers generally described their experiences of clinic consultations as a passive experience with little opportunity for participation and learning. Carers were particularly negative about their experiences with staff of mainstream health services in town E. Long waiting times and poor communication about upcoming specialist (e.g., paediatrician) and health service (e.g., allied health) visits in some communities were also noted as barriers by carers and service providers. Several carers, service providers and a healthcare practitioner discussed the importance of ensuring that community clinics were a clean and inviting, culturally secure space for carers and children to visit. 

Treatment acceptability for skin infections was discussed by all groups. The use of BPG injections for the treatment of skin sores was particularly contentious. Some parents and carers spoke of the pain associated with the injection and the traumatising effect it could have on their children, while others indicated a preference for BPG injections compared to a course of oral antibiotics. Carers indicated that treatment options for their child’s skin infection were generally not discussed during consultations. Furthermore, when creams or oral antibiotics were prescribed, carers reported often not receiving sufficient instruction on how and when to administer them.


*“Sometimes the parents get hurtful for that kid having the needle. They don’t want to see that kid screaming there from that needle. And that needle is big it’s not a little needle.”*
(carer)


*“I have to say in my previous experience, rarely [referring to a strong relationship between clinic staff and clients]. There’s been a few times where I’ve seen it, but rarely just because the turn-over of staff is so high really in those posts. There’s generally just one nurse out on their own and they have more than enough—to be on call for 12 hours a day. The turn-over, I guess, has meant that it’s been difficult for people to maintain those kinds of relationships with the community for an extended period of time.”*
(healthcare practitioner)


*“The health clinic doesn’t give them an option for treatment for their skin but it’s whatever the nurse on at the time wants. They don’t care. And that’s the problem.”*
(other service provider)

#### 3.2.3. At the System Level

System-level themes that recurred during interviews and group discussions included community clinic staff shortages and rapid staff turnover, the perceived organisational instability at the time of the study of the AMS servicing the region, a lack of communication and coordination between community services on child health matters, inefficient use of financial resources, and a perceived lack of AMS community engagement and outreach policies. The occasional occurrence of stock shortages in community clinics for medications and other medical supplies (including creams and antibiotics used for the treatment of skin infections) was also mentioned by some healthcare practitioners. The provision of free care for Aboriginal people in community AMS clinics and for children in mainstream health services in town was considered an important system-level enabler for health service utilisation.

The lack of trained Aboriginal Health Workers (AHWs) was highlighted by all three respondent groups as an important limitation to health service utilisation in the communities. Healthcare practitioners talked about how this hampered the planning and implementation of outreach activities, while carers and service providers commented on the absence of a genuine point of cultural interface between the community and the clinic. Several healthcare practitioners and service providers suggested that community pressures, expectations and perceived cultural limitations around community interactions may be barriers to the training and recruitment of Martu AHWs.


*“[in reference to mainstream health services in town E] We should have Aboriginal health workers because they will make sure everything is right because they have had so much primary health care.”*
(carer)


*“Local knowledge. They know who the people are. They know who to chase. They will also be able to tell me if people are here or not, if there’s someone I need to catch for one reason or another. And it’s a nice link, and they will tell you what’s culturally appropriate and not appropriate. So having a good health worker is important for the nurses, and I think if you have a clinic with high turnover of nursing staff, having a health worker as your local resource is fantastic.”*
(healthcare practitioner)


*“The problem is, I think, the clinic doesn’t talk that much with [name of community coordinator], the clinic doesn’t talk that much with us. So they’re very much isolated. […] But you can probably gather that communication is the key, and that’s what’s lacking.”*
(other service provider)

### 3.3. Need Factors: Aspects of Perceived and Evaluated Need for Health Service Utilisation

#### 3.3.1. At the Client Level

Service providers and healthcare practitioners generally highlighted the normalisation of skin infections as an important issue. Some parents and carers echoed this and indicated that there was insufficient knowledge amongst community members around skin infections and their impact on health. The three respondent groups saw this as a contributing factor to delayed health service presentations. Several service providers and healthcare practitioners suggested that seeking care for skin infections is often delayed until symptoms became more severe, causing discomfort, whether by pain or by itching. Some study participants also commented on how they believed carers might opt to self-treat skin sores, or use bush medicine, before seeking treatment through health services.


*“... she [her daughter] had a sore on her arm so I took her on a plane to Hedland [...] It looked like a blister it was black. I didn’t know it was any problem. The baby doctor said I had to go.”*
(carer)


*“I don’t know what they see as a normal part of life in their skin problems. I know that most likely they’re used to it. They’re used to, "Mom had heaps of boils when she was young, so why should I start running to the clinic with my child when they have them, because they’re just there anyway?"”*
(healthcare practitioner)


*“It’s not neglect, it’s not laziness, it’s just too vague. No one knows what it [the skin conditions] is.”*
(other service provider)

#### 3.3.2. At the Provider Level

Carers, service providers and healthcare practitioners all shared the opinion that the AMS provides insufficient education on health, including skin health. Some argued that carers would be more likely to visit the clinic for a skin infection if culturally appropriate education were prioritised and carers were made more aware of the importance of skin infections. This was also voiced strongly by parents and carers themselves. Some service providers talked about their own role in actively encouraging carers to take their child to the clinic when they noticed skin infections.


*“The health care in these communities is really poor because they don’t teach us.”*
(carer)


*“So it’s different ways of looking at it, but I think education towards the kids about their health needs to be developed and maintained in a different way. And I think the healthcare professions needs to step in and do it, and not rely on the teachers to do it, because the teachers have enough other things they need to educate the things in. I think by getting a new face in, explaining, showing, and then it might be one or two things the kids remember later on that might prevent one kid to have boils or scabies or something like that.”*
(healthcare practitioner)


*“If there’s something that I don’t like the look of, I just get the mom to take them to the clinic, or I do. The clinic’s literally next door to us, so it’s not an issue. It’s not hard taking them. It’s not a long trip or anything.”*
(other service provider)

#### 3.3.3. At the System Level

No need factors at the systems level were documented.

## 4. Discussion

Our explorative qualitative study has identified a wide range of factors that affect health service utilisation by Aboriginal people living in remote communities of the Pilbara region in Western Australia. While some of these relate specifically to childhood skin infections, the focal topic of our study, most describe enablers and barriers that are of broader relevance to rural and remote health. The key findings of our qualitative analysis are summarised in Table 6.

The need for establishing and maintaining trusting relationships between health services and the community was a recurrent theme amongst all participant groups. The retention of healthcare practitioners is a crucial factor in ensuring the staff stability required for such relationships to develop [24]. Many elements can affect the stability of the remote health workforce. In our study, healthcare practitioners spoke of stressors around understaffing, organisational instability, long working hours and the isolation that comes with working in remote communities. Similar findings have been reflected in other studies [37,38,39]. Other factors that have been associated with remote health workforce retention include the provision of training and professional development opportunities, good clinic infrastructure and comfortable housing conditions, competitive remuneration, and building a supportive work environment that recognises and rewards individual contributions [40]. Furthermore, actively encouraging healthcare practitioners to form relationships with community members can in itself be an important promoter of healthcare practitioner retention [41]. AMS organisations might therefore consider implementing policies and guidelines that facilitate the introduction of new healthcare practitioners to a community, which may help reduce staff turnover and improve health service utilisation. 

Our findings indicate a need for improved cultural competency [25] amongst rural and remote health service staff. This relates to the ability of non-Aboriginal healthcare practitioners to understand and engage with Aboriginal experiences of health and illness, but also pertains to recognising the detrimental effects that colonisation, prejudice and racism can have on health service delivery and use [24,42,43,44,45,46,47,48]. Comments from parents, carers and community service providers illustrated failings in this area and spoke to a need for a more inclusive patient-centred approach to health care and clinic consultations. Culturally responsive, patient-centred care requires a sensibility and self-reflexivity from non-Aboriginal healthcare providers to avoid further contributing to victim blaming and inciting feelings of shame [25,47,49], both of which have been shown to negatively impact on health service utilisation [48,50]. 

Community service providers, parents and carers referred to a perceived lack of communication and collaboration around child health in the communities. This included insufficient communication around visiting health services, clinic representatives not participating in community coordination meetings and a lack of community engagement through outreach and health promotion activities. Comments suggested that although such activities had been organised by the community clinic in the past, they were dependent on the initiative of certain individual healthcare practitioners. The activities would cease once these healthcare providers moved on, illustrating the need for structural solutions to ensure the continuity of community engagement and outreach activities.

An additional strategy to facilitate a positive health service experience for Aboriginal people is the employment of local AHWs [41,51,52,53]. AHWs fulfil the role of a primary healthcare worker, serve as a cultural broker that facilitates interaction between the health service and the community, and work in close partnership with non-Aboriginal healthcare practitioners to help assure culturally appropriate care [54,55]. Although AHWs had been trained and employed previously in our study setting, none were based at any of the community clinics at the time of our study. This shortage of AHWs was brought up by study participants. While our study participants pointed towards the possible challenges that AHWs might face in negotiating community and cultural roles and expectations on the one hand and health service expectations on the other, other challenges documented elsewhere include community pressures, inadequate training and career pathways, inequitable and inflexible working conditions and difficult working relationships [40,51,56]. The lack of AHWs is illustrative of the overall shortage of Aboriginal health professionals in Australia. While Aboriginal and Torres Strait Islander people make up 2.8% of the Australian population [57], only 0.5% of Australian medical practitioners and 1.1% of nurses and midwives identify as such [58,59]. Efforts to bridge this gap need to be increased.

We identified the pain and discomfort associated with BPG injections for the treatment of skin sores to be a factor that adversely affects health service utilisation for childhood skin infections. The administration of BPG is especially painful because of the high viscosity of the liquid and the large volume required (2.3 mL) to be injected into the thigh or buttock muscles [60]. Healthcare practitioners in our study acknowledged this barrier, and some discussed their reluctance in using the injection for treating children. However, considering its high effectiveness in treating skin sores and the perceived advantage of avoiding the need for a course of oral antibiotics, some healthcare providers and a few parents/carers preferred this treatment option overall. Our findings suggest that not all healthcare practitioners have received training on the administration of BPG injections and associated pain minimisation techniques. Ensuring such training and training resources [61] are provided to AMS clinical staff may prove beneficial. 

An additional factor that was perceived to hamper health service utilisation for childhood skin infections was their apparent normalisation in the remote Aboriginal community setting and an unawareness concerning their impact on overall health. In our study this was described mostly from the perspective that parents, carers and children did not recognise the significance of such infections, leading to delayed presentations. However, other studies have pointed out that normalisation of skin infections also occurs from the healthcare practitioner side, as is reflected in clinical guidelines that have set treatment thresholds for skin sores too high in the past [62] and a propensity to underdiagnose skin infections in the regional hospital setting [63]. These findings highlight the need to ensure parents, carers, healthcare practitioners and other community service providers are informed about the importance of skin health and the acute and long-term health risks posed by childhood skin infections.

Several limitations apply to the findings reported in this study. The researchers who conducted the study were non-Aboriginal and interpreted the data from their perspective. We performed informant feedback with Martu, other participant groups and senior Aboriginal and non-Aboriginal researchers to support the reliability of the information presented here. Our findings are reflective of attitudes and opinions held amongst healthcare practitioners and service providers present in the communities at the time of our data collection activities. Since this is a largely transient workforce, the views documented in our analysis are not necessarily representative for the study communities. Similarly, the relative importance of some barriers and enablers documented here might differ from those experienced in other communities. Nevertheless, we believe that our analytical framework and the findings that emerged from our analysis provide a useful lens through which to do similar research in other remote Aboriginal communities.

## 5. Conclusions

Our study outcomes illustrate how practices around health service utilisation for skin infections are the product of an interplay between many different factors at the client, health provider and health systems level. Public health strategies aimed at improving care-seeking practices in relation to skin health should aim to reflect this complexity in the interventions that they propose. Through its conceptualisation of need, enabling and predisposing factors, the Andersen model provides a framework around which the development of such a program could be structured. Following on from these three factors, the outcomes of our study suggest that health service utilisation for skin infections could be significantly improved in the study area by (i—need) denormalising skin infections in the community and amongst (health) service providers; (ii—enabling) creating the circumstances in which a trusting and patient-centred relationship can be formed between clients and healthcare practitioners, while also ensuring that healthcare practitioners are adequately trained in administrating BPG injections, are aware of pain minimisation techniques and offer informed options about the relative benefits of the different treatment strategies; and (iii—predisposing) improving cultural awareness among healthcare practitioners and encouraging them to actively engage with traditional knowledge and practices where appropriate and feasible. Since concluding this study, we have worked with community members, healthcare practitioners and an expert panel to develop community and healthcare practitioner-oriented local resource and national guidelines that address several of the identified barriers [64,65]. Furthermore, as a result of the research, a community engagement position was funded to help facilitate community members in seeking prompt treatment for skin and other conditions. Additional studies to inform community-based skin health programs in remote Aboriginal communities have commenced.

## Figures and Tables

**Figure 1 ijerph-17-00808-f001:**
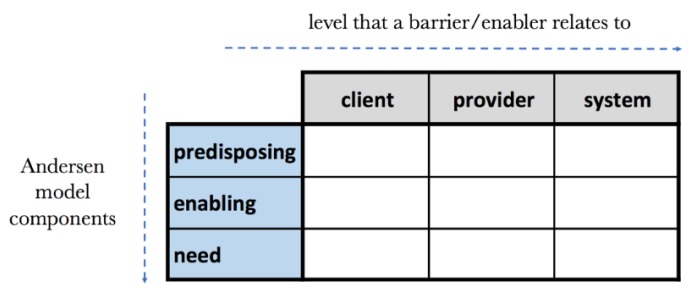
The health service utilisation matrix.

**Table 1 ijerph-17-00808-t001:** Description of study area localities.

	Aprox. Population	% Abl	AMS Clinic	GP Clinic	School	Shop	Police Station	Swimming Pool	Public Hospital	Road Access	Access by Air
Community A	250-500	76%	yes	no	yes	community store	yes	yes	no	unsealed	airstrip
Community B	100-200	91%	yes	no	yes	community store	no	no	no	unsealed	airstrip
Community C	100-200	88%	yes	no	yes	community store	no	no	no	unsealed	airstrip
Community D	<100	88%	yes	no	yes	community store	no	no	no	unsealed	airstrip
Town E	5000+	10%	no	yes	yes	supermarket	yes	yes	yes	sealed	airport

Abl = Aboriginal people; AMS = Aboriginal Medical Service; GP = general practitioner. Source: Australian census 2011.

**Table 2 ijerph-17-00808-t002:** Summary of data collection activities.

	Description of Participant Group	# of Interviews	# of FGDs	# of Participants		Sampling Method
				Total	Abl	
**Group 1 - parents/carers**	mothers, ’aunties’^, ’uncles’^ of young Aboriginal children	8	3	16	16	convenience & snowball
**Group 2 - healthcare practitioners***	remote area nurses, nurse practitioners, child and community health nurses, midwives	8	2	15	2	purposive
**Group 3 - other service providers**	teachers, Aboriginal education workers, early child care workers, community organisation staff	18	4	25	2	purposive

# = number; FGDs = focus group discussions; Abl = Aboriginal people. ^ In Aboriginal culture, the terms auntie and uncle are used to refer to an older person and connotes respect. * none were Aboriginal Health Workers or Aboriginal Health Providers (there were no active AHWs or AHPs based in the communities at the time of data collection).

**Table 3 ijerph-17-00808-t003:** Factors that affect health service utilisation in remote Pilbara communities according to parents/carers.

	Carer & Child (Client)	Clinic & Staff (Provider)	System
**Predisposing**	Shyness & shame* Traditional remedies & self-treatment * Negative past experiences with clinic* **Language barrier** **Fear of judgement (incl. DCP&FS)**	**prejudice**	
**Enabling**	Low costs associated with medical care* Good perception of clinic staff* Access to ’Closing the Gap’ benefits	Engaging & culturally secure staff & practices* Established relationship between staff & client* Clinic waiting time not too long* Comfortable, inviting clinic facility* Acceptability of treatment* **Clinic does outreach activities** **Clear communication re visiting health services** **Patient engagement**	Trained Aboriginal health workers* Ensuring adequate medical supplies
**Need**	Lacking awareness re skin infections* Delayed presentations & self-treatment* Normalisation of skin infections*	Clinic not providing sufficient health education*	

DCP&FS = Department of Child Protection and Family Support. * = theme was discussed by all three participant groups. Themes marked in bold indicate topics that were discussed by parents/carers and/or service providers, but not by healthcare providers.

**Table 4 ijerph-17-00808-t004:** Factors that affect health service utilisation in remote Pilbara communities according to healthcare practitioners.

	Carer & Child (Client)	Clinic & Staff (Provider)	System
**Predisposing**	Shyness & shame* Traditional remedies & self-treatment* Negative past experiences with clinic* Apathy & disempowerment Priorities, values & norms - health & child rearing Cultural taboos Tolerance for discomfort & pain Lacking (health) education	Stressors associated with work environment Jadedness/frustration Training/knowledge gaps	Reconciling traditional and modern medicine
**Enabling**	Low costs associated with medical care* Good perception of clinic staff* Access to ’Closing the Gap’ benefits	Engaging & culturally secure staff & practices* Established relationship between staff & client* Clinic waiting time not too long* Comfortable, inviting clinic facility* Acceptability of treatment*	Trained Aboriginal health workers* Ensuring adequate medical supplies Free or low-cost medical care Adequate staff levels & low turnover AMS governance & stability Efficient use of resources Good collaboration with other services Community engagement & outreach policies
**Need**	Lacking awareness re skin infections* Delayed presentations & self-treatment* Normalisation of skin infections*	Clinic not providing sufficient health education*	

AMS = Aboriginal Medical Service. * = theme was discussed by all three participant groups.

**Table 5 ijerph-17-00808-t005:** Factors that affect health service utilisation in remote Pilbara communities according to other service providers.

	Carer & Child (Client)	Clinic & Staff (Provider)	System
**Predisposing**	Shyness & shame* Traditional remedies & self-treatment* Negative past experiences with clinic* **Language barrier** **Fear of judgement (incl. DCP&FS)** Apathy & disempowerment Priorities, values & norms - health & child rearing Cultural taboos Tolerance for discomfort & pain Lacking (health) education	Stressors associated with work environment Training/knowledge gaps	Reconciling traditional and modern medicine
**Enabling**	Low costs associated with medical care* Good perception of clinic staff* **Carer is healthy**	Engaging & culturally secure staff & practices* Established relationship between staff & client* Clinic waiting time not too long* Comfortable, inviting clinic facility* Acceptability of treatment* **Clinic does outreach activities** **Clear communication re visiting health services** **Patient engagement**	Trained Aboriginal health workers* Free or low-cost medical care Adequate staff levels & low turnover AMS governance & stability Efficient use of resources Good collaboration with other services Community engagement & outreach policies
**Need**	Lacking awareness re skin infections* Delayed presentations & self-treatment* Normalisation of skin infections*	Clinic not providing sufficient health education* **Actively encourage child/carer to go to clinic**	

DCP&FS = Department of Child Protection and Family Support; AMS = Aboriginal Medical Service. * = theme was discussed by all three participant groups. Themes marked in bold indicate topics that were discussed by parents/carers and/or service providers, but not by healthcare provider.

**Table 6 ijerph-17-00808-t006:** Summary of factors that facilitate health service utilisation and associated themes.

**General**	
The importance of establishing a relationship between healthcare practitioners and parents/carers.	
*client:*	good perception of clinic staff.
*provider:*	established relationship between staff and clients; stressors associated with work environment; jadedness/frustration.
*system:*	adequate staff levels and low turnover; AMS governance and stability.
The need for the active engagement of parents/carers in their health care through culturally appropriate practice.	
*client:*	shyness and shame; language barrier; fear of judgement; cultural taboos.
*provider:*	prejudice; engaging and culturally secure staff and practices; patient engagement; training/knowledge gaps.
*system:*	trained Aboriginal health workers; reconciling traditional and modern medicine; AMS governance and stability.
The need for cross-organisational communication and collaboration around child health	
*provider:*	clear communication re visiting health services.
*system:*	efficient use of resources; good collaboration with other services; community engagement and outreach policies.
**For Skin Infections**	
The need to address normalisation and provide parent/carer education on the importance of skin health and skin infections.	
*client:*	need for (health) education; increasing awareness and denormalisation of skin infections; timely presentation to a clinic; acknowledging discomfort and pain.
*provider:*	clinic provides sufficient health education; actively encourage child/carer to go to clinic.
Negative experiences associated with BPG injections	
*client:*	ensure positive experiences at the clinic.
*provider:*	acceptability of treatment; ensure clinic staff are trained to administer BPG injections as painlessly as possible.

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
