# Peer review of "Barriers and Enablers of Health Service Utilisation for Childhood Skin Infections in Remote Aboriginal Communities of Western Australia"

_ijerph, 2020, doi:10.3390/ijerph17030808_

Round 1

Reviewer 1 Report

Thank you for this interesting qualitative research exploring barriers to accessing primary health services for skin infections.

Section 2.1 (line 57) and Section 2.4:

Thank you for specifying that there was substantial involvement of senior Aboriginal researchers in reference groups guiding this study. 

1) Do you have the express consent of appropriately senior Martu representatives to name them as the group involved in the study?

2) Do the benefits of specifically naming the Martu people outweigh the potential risks eg. Shame or embarrassment relating to skin health, offence caused by negative comments in the results section from some participants (eg. Martu clients have poor health literacy, language skills or are apathetic about skin health) (line 183)?

Section 3 Results:

3) Results from the clients themselves seem lost in the results, and seem a little drowned out by other comments ‘speaking for’ and ‘speaking about’ clients. Some comments appear to use negative stereotyping and race-based assumptions (eg. Martu clients are apathetic, Martu children can inherently tolerate pain better - which were appropriately challenged in the discussion). Were these comments from Martu clients themselves or from other people?  Can the results be reframed in a way that better clarifies where these perceptions or assertions come from and allow greater clarity of what clients themselves say?

Section 3 line 155

4) “Few healthcare practitioners and service providers also spoke on this topic.” This sentence is a little ambiguous. Did you mean “Few healthcare practitioners and service providers spoke on this topic” or “A few healthcare practitioners and service providers also spoke on this topic”

Author Response

REVIEWER 1

Thank you for this interesting qualitative research exploring barriers to accessing primary health services for skin infections.

Section 2.1 (line 57) and Section 2.4:

Thank you for specifying that there was substantial involvement of senior Aboriginal researchers in reference groups guiding this study. 

1) Do you have the express consent of appropriately senior Martu representatives to name them as the group involved in the study?

2) Do the benefits of specifically naming the Martu people outweigh the potential risks eg. Shame or embarrassment relating to skin health, offence caused by negative comments in the results section from some participants (eg. Martu clients have poor health literacy, language skills or are apathetic about skin health) (line 183)?

Thanks for these two closely linked comments, as we agree with the reviewer that these type of questions are important to fully consider and constitute some of the reasons why it was important to frame this study within a reference group of senior Aboriginal researchers and include senior community leaders in the review and authorship of this paper.  Co-author Margaret Samson is a senior Martu community chairperson in Jigalong, one of the communities that was the focus of this study.  Co-author Robby Chibawe is the CEO of the Puntukurnu Aboriginal Medical Service, which is led by a board of senior Martu people.  Both co-authors have reviewed and approved the submitted version of the paper, and agree that including mention of the Martu in the paper is of overall benefit.

In naming them, we wanted to underline that the Martu are a distinct Aboriginal group in the Western Desert who experience layers of disadvantage that are tied to their unique relation to other groups and their remoteness.  We wanted to be able to emphasise their distinctive characteristics and barriers presented to them, and the need for health services to understand those specificities, which we hope would be of benefit to service providers working in Martu communities.  We debated whether naming the communities in the paper would be appropriate but refrained from doing so for reasons suggested by the reviewer.  We felt that referring instead to the Martu would strike an appropriate benefit/risk balance in this way.  We have followed a similar line in other papers we have published on our work in this region.

We are pleased that it was apparent to the reviewer (see their comment 3) that our intent was to challenge negative stereotyping and try to frame these types of comments appropriately from a strength-based perspective.  We also wanted to signal to the reader (in a statement that prefaces the annex with structured quotes) that such comments do not reflect the views of the authors, but that we have nevertheless strived to provide a balanced and honest representation of the views expressed -both positive and negative- by all participant groups.  We would also like to inform the reviewer that we are currently preparing a separate paper based on this and other studies in the region to more thoroughly point towards and discuss some of the prejudiced, discriminatory and colonial attitudes that we have encountered in our work in the region.

Section 3 Results:

3) Results from the clients themselves seem lost in the results, and seem a little drowned out by other comments ‘speaking for’ and ‘speaking about’ clients. Some comments appear to use negative stereotyping and race-based assumptions (eg. Martu clients are apathetic, Martu children can inherently tolerate pain better - which were appropriately challenged in the discussion). Were these comments from Martu clients themselves or from other people?  Can the results be reframed in a way that better clarifies where these perceptions or assertions come from and allow greater clarity of what clients themselves say?

We agree with the reviewer that this is another important point to consider in writing this type of paper.  This observation is a reflection of how the study was conceived around three distinct participant groups (Martu parents and carers, healthcare practitioners, other service providers), meaning that in a quantitative sense we ended up with more qualitative data from participant groups who ‘spoke about’ clients than from the one participant group that ‘spoke for’ clients.  We have now included a mention of this in the limitations section. 

Nevertheless, we have taken care to try and privilege and emphasise the important implications of what parents and carers have shared with us, particularly in our overall discussion of the results.  We also purposely set out to privilege the perspectives of Martu parents and carers in the results by describing them first in the results section when reporting on themes that were discussed by that participant group (not all themes were brought up by all participant groups, as illustrated in tables A3 to A5) and by structuring quotes in the annex (and those now added in the manuscript) in a way that those from Martu parents and carers appear first.  We intended this to help reinforce how Martu parents and carers feel about the health service experience and frame the ensuing comments from other participant groups.  

In response to the above comment we have also reviewed the results and have edited them to further improve clarity in terms of which assertions were made by which participant group.  We have also included a selection of key quotes in the results section to this end, as was suggested by reviewer 2. 

Section 3 line 155

4) “Few healthcare practitioners and service providers also spoke on this topic.” This sentence is a little ambiguous. Did you mean “Few healthcare practitioners and service providers spoke on this topic” or “A few healthcare practitioners and service providers also spoke on this topic”

This should indeed have read “A few […]”.  Corrected.

Reviewer 2 Report

This is a beneficial piece of work with wider implications regarding Aboriginal barriers/enablers to health service use beyond just skin infections.

I think overall key quotes relating to the themes being discussed should be dispersed throughout the article to support statements that have been made rather than compiled in a 30+ page supplementary file. I’m just not sure about the degree to which the sup file will be accessed and don’t want such rich data to be missed. At minimum if the supplementary file is kept, it should at least be shortened (i.e. for Shame you have 4 quotes from each of the 3 groups, I think 1 -2 quotes per group is more than sufficient to support each theme).

Below are some minor revisions to be considered:

Paragraph 2 Introduction - also worth mentioning the disproportionality high rate of homeless/overcrowding amongst Aboriginal people (x10 more likely to be homeless and x16 more likely to be living in severely overcrowded dwellings). I would assume overcrowding would be a risk for skin infection and thus probably worth mentioning.   https://www.aihw.gov.au/reports/housing-assistance/indigenous-people-focus-housing-homelessness/contents/at-a-glance

Line 87 –not sure “at least” required i.e. ‘all healthcare practitioners had several years’ of experience’ is sufficient

Section 3.2.2 – curious if it was noted that the actual length of appointment was a barrier at all? Given an earlier comment about English not the first language for majority of the carers, wondering if this contributes to the negative experience- i.e. feeling rushed and not listened to, also could account for why treatment appears not to be discussed within the consultation.

Lines 346 -  healthcare instead of health care?  Seems you have used “health care” to describe provision of services and “healthcare” for system/workers, if so I think this one needs to be changed for consistency.

Tables A3 – A5 –Is it possible to have the content from these 3 tables in one table? As is it’s a bit difficult to read and compare themes across the three groups. Two suggestions to improve readability are three rows per each of the current rows (i.e. for predisposing you could then have a healthcare, service and parent row), or alternatively keep the current rows and turn more into a figure where the answers are colours and a key for responses (i.e. red for all three, blue for parent + service, green for parent only etc..)

Box A6 – This doesn’t necessarily need to be addressed, but the way general and skin infections are presented are inconsistent in the language used. Under “general” the things that would influence positive health service use are discussed (i.e. low turnover, good relationships, being culturally sensitive etc.) however for “skin infections” these are all referred to in the negative/limiting tense (lack of xxx, delayed presentation, not providing education  etc..). Given that the high turnover of staff and lack of governance etc. is what is discussed in the text should that “general” not align with those comments rather than the ideal model of care? Or alternatively the “skin infection” comments reversed so that it reflects the ideal model (i.e. provision of education, de-normalising skin infections etc.).

Author Response

This is a beneficial piece of work with wider implications regarding Aboriginal barriers/enablers to health service use beyond just skin infections.

I think overall key quotes relating to the themes being discussed should be dispersed throughout the article to support statements that have been made rather than compiled in a 30+ page supplementary file. I’m just not sure about the degree to which the sup file will be accessed and don’t want such rich data to be missed. At minimum if the supplementary file is kept, it should at least be shortened (i.e. for Shame you have 4 quotes from each of the 3 groups, I think 1 -2 quotes per group is more than sufficient to support each theme).

Thanks for this comment, as it is something we had discussed prior to submission, and so appreciate the reviewer’s thoughts on this. We have now integrated a selection of quotes (one per respondent group) into every results section as to give readers a more direct impression of what the various respondent groups said.  We still refer the reader to the supplementary file in the first results paragraph, as we want to give the reader the option to explore additional quotes, but we have significantly shorted the document as to make it more easily navigable (24 pages).

Below are some minor revisions to be considered:

Paragraph 2 Introduction - also worth mentioning the disproportionality high rate of homeless/overcrowding amongst Aboriginal people (x10 more likely to be homeless and x16 more likely to be living in severely overcrowded dwellings). I would assume overcrowding would be a risk for skin infection and thus probably worth mentioning.   https://www.aihw.gov.au/reports/housing-assistance/indigenous-people-focus-housing-homelessness/contents/at-a-glance

We agree with the reviewer that the factor of overcrowding is indeed important to point out in the introduction section.  With thanks for the reference.  Introduction section updated accordingly.

Line 87 –not sure “at least” required i.e. ‘all healthcare practitioners had several years’ of experience’ is sufficient

There was quite some variation in terms of the duration healthcare practitioners involved in the study had been working in the field of remote Aboriginal health.  The majority we spoke to had more than 5 years of experience, some decades.  A couple only had a few years of experience. We would prefer leaving in “at least”, as without it the sentence might wrongfully convey an interpretation that we only included relatively inexperienced healthcare practitioners in our study.

Section 3.2.2 – curious if it was noted that the actual length of appointment was a barrier at all? Given an earlier comment about English not the first language for majority of the carers, wondering if this contributes to the negative experience- i.e. feeling rushed and not listened to, also could account for why treatment appears not to be discussed within the consultation.

This is indeed an interesting point.  Review of the transcripts did not reveal direct mention of the length of consultations as a potential barrier, although some of the other barriers described in the article (as suggested by the reviewer) do indeed at least suggest that inadequate consultation lengths may be an intrinsic factor.  Unfortunately the question was not asked and we do not have data to support a statement on this.

Lines 346 -  healthcare instead of health care?  Seems you have used “health care” to describe provision of services and “healthcare” for system/workers, if so I think this one needs to be changed for consistency.

That is correct.  We have corrected this, as suggested.

Tables A3 – A5 –Is it possible to have the content from these 3 tables in one table? As is it’s a bit difficult to read and compare themes across the three groups. Two suggestions to improve readability are three rows per each of the current rows (i.e. for predisposing you could then have a healthcare, service and parent row), or alternatively keep the current rows and turn more into a figure where the answers are colours and a key for responses (i.e. red for all three, blue for parent + service, green for parent only etc..)

We appreciate the reviewer’s thoughts on this, as the summary of themes across all participant groups in a succinct yet comprehensive manner was indeed a challenge given the extent of topics covered.  We did in fact try a format similar to the one suggested by the reviewer, where we tried to present all themes for all participant groups in one table.  After some experimentation we decided on the current multi-table format, using asterisks and bold font to highlight cross-participant group similarities and differences.  Having explored different options, we would prefer to stay with the current multi-table format.

Box A6 – This doesn’t necessarily need to be addressed, but the way general and skin infections are presented are inconsistent in the language used. Under “general” the things that would influence positive health service use are discussed (i.e. low turnover, good relationships, being culturally sensitive etc.) however for “skin infections” these are all referred to in the negative/limiting tense (lack of xxx, delayed presentation, not providing education  etc..). Given that the high turnover of staff and lack of governance etc. is what is discussed in the text should that “general” not align with those comments rather than the ideal model of care? Or alternatively the “skin infection” comments reversed so that it reflects the ideal model (i.e. provision of education, de-normalising skin infections etc.).

Thanks for pointing this out, as it certainly was not intentional.  We have edited the ‘skin infection’ section of the box to correspond with the language used in the ‘general’ section.